

# Using GRACE to derive corrections to precipitation data sets and improve modelled snow mass at high latitudes

Emma L. Robinson[1] and Douglas B. Clark[1]

[1]Centre for Ecology & Hydrology, Maclean Building, Benson Lane, Crowmarsh Gifford, Wallingford OX10 8BB

**Correspondence:** Emma L. Robinson (emrobi@ceh.ac.uk)

**Abstract.** The amount of lying snow calculated by a land surface model depends in part on the amount of snowfall in the meteorological data that are used to drive the model. We show that commonly-used data sets differ in the amount of snowfall, and more generally precipitation, over four large Arctic basins. An independent estimate of the cold season precipitation is obtained by combining water balance information from the Gravity Recovery and Climate Experiment (GRACE) with estimates

of evaporation and river discharge, and is generally higher than that estimated by four commonly-used meteorological data sets. We use the Joint UK Land Environment Simulator (JULES) land surface model to calculate the snow water equivalent (SWE) over the four basins. The modelled seasonal maximum SWE is 38% less than observation-based estimates on average and the modelled basin discharge is significantly underestimated, consistent with the lack of snowfall. We use the GRACE-derived estimate of precipitation to define per-basin scale factors that are applied to the driving data and increase the amount of cold

season precipitation by 28% on average. In turn this increases the modelled seasonal maximum SWE by 30%, although this is still underestimated compared to observations by 19% on average. A correction for undercatch of precipitation by gauges is compared with the the GRACE-derived correction. Undercatch correction increases the amount of cold season precipitation by 23% on average, which indicates that some, but not all, of the underestimation can be removed by implementing existing undercatch correction algorithms. However, even undercatch-corrected data sets contain less precipitation than the GRACE-

derived estimate in some regions, and it is likely that there are other biases that that are not currently accounted for in gridded meteorological data sets. This study shows that revised estimates of precipitation can lead to improved modelling of SWE, but much more modest improvements are found in modelled river discharge. By providing methods to better define the precipitation inputs to the system, the current study paves the way for subsequent work on key hydrological processes in high-latitude basins.

## 1 Introduction

Seasonal snow cover is an important part of the hydrological cycle over a large part of the Northern Hemisphere, with approximately 45 million km$^2$ of the land (excluding Greenland) covered by snow at the seasonal maximum (Mudryk et al., 2014). Snow plays an important role in the energy, water and biogeochemical cycles of these areas. The accumulation of water in the snow pack until the spring thaw dominates the seasonal cycle of runoff and river flow in many northern river basins (Grabs et al., 2000) and the physical properties of the snow pack, such as its high albedo, can lead to strong links between the snow

pack and the overlying atmosphere (Cohen and Rind, 1991; Mote, 2008).





Many studies have documented changes in seasonal snow cover in recent decades. The extent of Northern Hemisphere snow cover decreased through most of the last century, with a larger rate of decrease since 1970 (Brown and Robinson, 2011). Correlation between changes in spring snow extent and air temperature (Brown and Robinson, 2011) reveal the effects of the snow-albedo feedback (Fernandes et al., 2009).

Observed changes in snow water equivalent (SWE) show more variation between regions. The seasonal maximum SWE has increased over northern Eurasia over 1966–2010 (Bulygina et al., 2011) but springtime SWE over the mountains of western North America has decreased over 1960–2002 (Mote, 2006).

Overall the picture is one of considerable variation with the years and season of analysis, locations and snow measure all being important (Brown and Mote, 2009), but with declines in snow more common, particularly at warmer or lower locations
(see Fig. 4.21 in Vaughan et al., 2013).

These analyses of historical snow cover have used a variety of data sources, principally ground-based point observations of precipitation and snow cover, and products derived from satellite data, each of which has strengths and weaknesses. Direct measurements of snowfall using gauges and of characteristics of the snow pack (e.g. SWE) are difficult to scale up to large areas for reasons including the high spatial variability in these quantities and the sparse network of observing stations (e.g.
Liston and Hiemstra, 2011). Inconsistent trends in precipitation and runoff in large northern catchments confirm that these data are of uncertain quality (Berezovskaya et al., 2004; Pavelsky and Smith, 2006). Satellite data can be used for recent decades, particularly to determine the presence or absence of snow cover. Passive microwave data from satellites can be used to assess SWE (e.g. GlobSnow; Takala et al., 2011) but the need for an algorithm to convert the measured radiance to SWE, coupled with the requirements for supporting data such as ground-based observations of meteorology, introduces uncertainty into the
final estimates.

An alternative to these observation-based methods is to use a land surface model (LSM) to describe and understand the evolution of the snow pack (e.g. Liston and Hiemstra, 2011). These models come with their own uncertainties and requirements for input data, but have the advantage of also being able to estimate future changes in snow cover as LSMs are also used as the land surface components of the climate models that are used to study future climates. Simulations using climate models indicate
that Northern Hemisphere snow cover extent will decrease in the future (Brutel-Vuilmet et al., 2013). Although warming will reduce the length of the snow-cover season, precipitation is predicted to increase over many mid- to high-latitude areas, meaning the impact on SWE may differ between regions (Räisänen, 2008). Any such changes in snowfall and snow cover will impact the wider hydrological cycle in these areas.

If we are to have confidence in these analyses of historical and possible future conditions from LSMs, we must first establish
that they can accurately represent the snow pack and related processes. To this end community intercomparison experiments have quantified the performance of land surface and other models in a variety of settings, some with a focus on snow processes and others looking at the wider hydrological cycle (e.g. Bowling et al., 2003; Essery et al., 2009; Haddeland et al., 2011). A recurring theme across many of these has been the uncertainty that stems from the input precipitation data, particularly in regions with seasonal snow cover.





This theme is also evident in snow simulations that use the Joint UK Land Environment Simulator (JULES) land surface model (Best et al., 2011; Clark et al., 2011). Burke et al. (2013) noted that that seasonal maximum SWE in pan-Arctic simulations of JULES was often underestimated in comparison to GlobSnow. This is consistent with the fact that the snowfall specified in the meteorological data that are used to drive JULES and other LSMs is often considerably less than the SWE

estimated by GlobSnow (Hancock et al., 2014). Ménard et al. (2015) suggest that the largest uncertainty in estimates of SWE from JULES arises from uncertainty in the input precipitation data.

A larger body of research concerns the ability of models to represent other aspects of land hydrology, particularly runoff and river flow, because measurements of river flow are generally considered more accurate than those of catchment-average precipitation and snow pack (Troy et al., 2011). Again a recurring finding of this work is the need to supply the models with

high quality precipitation data and the uncertainty in these data (Decharme and Douville, 2006; Fiedler and Döll, 2007; Tian et al., 2007; Biemans et al., 2009; Wisser et al., 2010; Weiland et al., 2015; Islam and Déry, 2017; Casson et al., 2018).

Furthermore, precipitation data are not only uncertain but possibly systematically biased low with studies suggesting that insufficient modelled river flow in high latitude catchments is a result of insufficient precipitation in the driving meteorology (Tian et al., 2007; Biemans et al., 2009; Alkama et al., 2010). It should be noted that these model biases are not limited to

land surface models (those that can be coupled to atmospheric models) but are also shown by global hydrology models, which typically focus on water resources and lateral fluxes (Haddeland et al., 2011).

Estimates of precipitation in areas with significant amounts of snow are made particularly difficult by the sparsity of the gauge network in many locations, the difficulty of obtaining an areal average in mountainous terrain, and the need to correct gauge measurements for undercatch and losses (Serreze et al., 2003; Adam and Lettenmaier, 2003; Adam et al., 2006). Cor-

recting for undercatch requires information on (at least) the equipment used, wind speed, and the phase of precipitation, and is itself a difficult and uncertain task (Serreze et al., 2003). Adam and Lettenmaier (2003) derived monthly correction factors for wind-induced undercatch and wetting loss that resulted in a 16% increase in DJF precipitation at the global scale, with much higher values in some regions. Wind-induced undercatch of solid precipitation was the largest source of bias. For stations north of 45° N Yang et al. (2005) calculated monthly correction factors of 80–120% of winter precipitation winter due to snowfall

and higher wind speeds. Correcting for orographic effects resulted in 6% higher global precipitation, 20% in orographically-influenced areas, using a water balance approach (Adam et al., 2006). Although these correction factors are substantial they are themselves uncertain and often only apply to a particular time period and data set, and some widely-used data sets do not include any correction factors (e.g. CRUNCEP Viovy, 2014).

Given the difficulties associated with gauge-based data, other products use information from satellite-based earth observation

or from atmospheric reanalyses, either alone or in combination with gauge data. However these approaches introduce their own uncertainties, such as how to relate the measured radiance to a precipitation rate, and considerable uncertainty remains (Stephens and Kummerow, 2007; Beck et al., 2017).

Given that areal precipitation is difficult to estimate directly using the aforementioned methods, an alternative approach instead seeks to infer precipitation from analysis of the total water storage (TWS) of the land surface as estimated by the Gravity

Recovery and Climate Experiment (GRACE) mission (Swenson, 2012). GRACE uses gravimetry to assess the changing mass



of the surface of the Earth, which can be interpreted as changes in water storage, revealing the dynamics of TWS at the surface on a monthly time scale. While the monthly changes in TWS can be observed, GRACE does not indicate what fluxes or components are the cause of the changes. However, by using the GRACE TWS anomalies, combined with observed or modelled estimates of river flow and gridded evaporation fluxes, it is possible to estimate the areal precipitation (Swenson, 2010).

Previous studies have used GRACE TWS to estimate precipitation inputs in boreal regions (Swenson, 2010; Seo et al., 2010; Behrangi et al., 2016) and cold mountainous basins (Behrangi et al., 2017). These have identified deficiencies or uncertainties in traditional estimates of precipitation, but with magnitude that varies between products, locations and times. Swenson (2010) compared GRACE-derived precipitation with two existing precipitation data sets and found a varied picture, with more variability in North America, while Behrangi et al. (2017) many precipitation products are significantly lower than the GRACE-derived estimate in cold mountainous endorrheic basins. Behrangi et al. (2016) compared several raw reanalysis outputs, as well as some satellite and gauge based products, and only found deficiencies in gauge-based data sets in some regions. These studies have all used LSM or reanalysis output to provide at least some of the ancillary data required to calculate precipitation from GRACE TWS. Since, in this study, the intention is to investigate how changes to input precipitation can result in changes to the LSM output, independent estimates of evapotranspiration and river flow have been used in preference to LSM output.

The remainder of this study investigates the hypothesis that many precipitation data sets that are used to drive land surface models contain insufficient cold season precipitation in mid- to high-latitude land areas, and that this leads to underestimation of SWE and river flow. Cold-season precipitation data are assessed against an estimate based on GRACE TWS data combined with independent estimates of evaporation and basin discharge. The results confirm that the precipitation data sets used to drive LSMs underestimate cold season precipitation in boreal regions. The GRACE-derived precipitation estimates are used to rescale the cold season precipitation in the data sets used to drive the JULES model, and this improves the representation of SWE, with a limited improvement in modelled river discharge.

In Sect. 2 the data sets used to carry out this study are described and in Sect. 3 the set up of the JULES model and the experiments carried out are defined. The method by which precipitation is calculated from GRACE TWS is described in Sect. 4. The JULES model runs are described and evaluated in Sect. 5 and the implications are discussed in Sect. 6.

## 2   Data sets

All analyses in this paper are carried out using data on a regular latitude-longitude grid at 1° resolution. Data with other resolutions are first regridded to this common grid with a modified version of SCRIP (Jones, 1998), using the conservative remapping method normalised by destination area fraction (Jones, 1999). This study focuses on the four major Arctic basins — the Ob, Yenisei, Lena and Mackenzie — which together account for approximately 68% of the total discharge into the Arctic Ocean (Grabs et al., 2000) and are marked by $O$, $Y$, $L$ and $M$ respectively in Fig. 1. The basins are defined using the 1° TRIP river network data of Oki and Sud (1998). For consistency with the observations of river flow (see Sect. 2.3), only the part of each basin that drains to the flow gauging station was considered when calculating spatial averages of observed or





modelled grids. The data sets used all vary in their temporal coverage; for consistency in analysis a "common overlap period" of 2002 – 2010 inclusive was used, during which time all of the driving data sets and the data sets used for the water balance precipitation estimates were available. In order to capture full annual cycles of snow accumulation and melt, annual averages were calculated from September 1st of one year to August 31st of the next, so the common overlap period contains eight

full years for analysis. All comparisons were carried out using data from this common overlap period, and averaged over the defined basins, unless explicitly noted otherwise.

## 2.1 Meteorological data

A key requirement of land surface models is a time series of meteorological data. To this end, several "driving" data sets have been created that represent realistic global climate for the recent past. These are generally based on global meteorological

reanalyses which have been bias-corrected to observational data sets. They provide a full set of the meteorological variables required to run an LSM, including air temperature, pressure and humidity, wind speed, incoming long- and shortwave radiation, as well as the precipitation with which this study is concerned.

Four meteorological data sets were used to drive the JULES model in this study. These are summarised in Table 1 and described below, with a focus on the precipitation data. The data were available at $0.5°$ resolution unless stated otherwise.

Basin mean monthly climatologies of all of the driving variables can be seen in Figs. S1 and S2.

The CRUNCEP v4 data set (CRUNCEP; Viovy, 2014) uses the CRU TS3.21 monthly data (Harris et al., 2013), including gauge-based precipitation, to bias correct the NCEP/NCAR Reanalysis 1 (Kalnay et al., 1996). The reanalysis is spatially interpolated from its original $2.5°$ resolution to the required $0.5°$ resolution with no adjustment for grid box elevation. No undercatch correction is applied to the gauge data and the total precipitation is provided (rather than separate estimates of

rainfall and snowfall).

The Princeton Global Forcing data v2 (PGF; Sheffield et al., 2006) is based on the NCEP/NCAR Reanalysis 1 with precipitation totals scaled to CRU TS monthly values and further statistical corrections applied to correct the number of rain days and to to scale to the final $1°$ and 3-hourly resolutions. Again total precipitation is provided. The reanalysis is spatially interpolated, including a correction for differences in grid box elevation. The version used in this study (v2) was *not* corrected for undercatch

(Justin Sheffield, personal communication, 2018) contrary to the description of the original data set in Sheffield et al. (2006).

Two variants of the WATCH Forcing Data methodology applied to ERA-Interim Data (WFDEI; Weedon et al., 2014) were used, differing only in the precipitation data used for bias correction: WFDEI-CRU uses CRU TS 3.1.01 until 2009 and CRU TS 3.21 from 2010 onwards (Harris et al., 2013; Trenberth et al., 2013), while WFDEI-GPCC uses GPCC v5 until 2009 and v6 from 2010 onwards(Schneider et al., 2014) and includes many more stations than CRU. Both variants apply undercatch

corrections (Adam and Lettenmaier, 2003) and use the ERA-Interim reanalyses (Dee et al., 2011). Variables other than precipitation are bias-corrected to CRU TS 3.1 until 2009 and CRU TS 3.21 from 2010 onwards in both versions of WFDEI. The reanalysis was spatially interpolated from its native N128 resolution (around $0.7°$ at the equator) to $0.5°$ resolution, including adjustment for differences in grid box elevation. Rainfall and snowfall were diagnosed separately in the reanalysis, with the fraction adjusted appropriately where differences in grid box elevation resulted in inappropriate precipitation phase.



All of these data sets have been widely used to drive land surface models in previous studies. All of them scale precipitation using data from CRU or GPCC, while an important difference is that only two (WFDEI-CRU and WFDEI-GPCC) include an undercatch correction. The undercatch correction implemented in Adam and Lettenmaier (2003) resulted in an increase in global precipitation of 12%, with an increase of 95% in some boreal regions. Thus this can be an important source of

underestimation of precipitation. Although the driving data sets are primarily used by land surface modellers, they include information from more widely-used precipitation data sets, and their basin averaged annual precipitation values are similar to those from a range of other data sets (Fig. S3).

## 2.2  GRACE

The twin GRACE satellites made detailed measurements of changes in Earth's gravity field from April 2002 – January 2017.

GRACE-Tellus used these to provide estimates of the TWS anomaly relative to the baseline mean over the years 2004–2009 (Swenson and Wahr, 2006; Swenson, 2012; Landerer and Swenson, 2012). GRACE may be used to study the dynamics of the water cycle on a monthly time step. Although it may not be used to estimate absolute stores, it gives an estimate of the integrated water fluxes into and out of a region.

Three methods have been developed to solve the gravity fields using spherical harmonics (CSR, GFZ, JPL). These are then

used to calculate the gridded TWS, so that three products are available. As is recommended by Sakumura et al. (2014), this study uses the ensemble average of the three solutions, which effectively reduces the uncertainty in the GRACE product.

The gravimetry measured by the GRACE satellites is converted to a TWS anomaly, making use of model output. Corrections for changes in atmospheric pressure are based on ECMWF analysis data, and corrections for post-glacial rebound are also modelled (Wahr et al., 1998). Destriping, Gaussian and degree 60 filters are applied to the gridded data. In order to correct for

signal attenuation introduced by the filters applied when calculating the grids from the spherical harmonics, the three TWS land grids must be multiplied by a gridded gain factor, which is derived by applying the same filters to the output from the Community Land Model 4 (CLM4; Landerer and Swenson, 2012).

Although the filters remove a significant component of spatially correlated and random errors, the gridded GRACE data are not independent of their neighbours. Uncertainties in the GRACE TWS data are generally highest at low latitudes, and lower

towards the poles. In the boreal regions in this study, estimated measurement error is around 15 mm (Landerer and Swenson, 2012). The application of the gridded gain factor means that the gridded GRACE land data cannot be used to study long-term trends (Landerer and Swenson, 2012).

The data used in this study were the GRACE RL05 gridded land TWS (Swenson, 2012; NASA, 2017). The data are available for the period April 2002 to January 2017. This study uses the fourteen years from September 2002 to August 2016, so that

analyses are carried out over entire cold seasons.

## 2.3  GRDC river discharge

Monthly river discharge data were take from the Global Runoff Data Centre Reference Dataset (GRDC, 2014). For each basin in the current study, the station closest to the river mouth (as defined in the routing grid, see Sect. 3) is used to define basin





discharge. In this study, basin discharge is quoted in units of mm, for consistency with the other variables, calculated by dividing the total discharge by the area of the basin draining to the station. There are missing observations at each station, and some stations ceased contributing data before the end of the study period. Months with missing data were filled using a mean-monthly climatology, calculated using all available data at the station. Data availability varies between basins, with

observations ending between 2003 and 2012. A summary of the available and filled data is shown in Table S1.

## 2.4  GLEAM v3.1a Evapotranspiration

The Global Land Evaporation Amsterdam Model v3.1a (GLEAM) provides estimates of evapotranspiration on a 0.25° resolution global grid, for the years 1980–2016, based on reanalysis and observational products (Miralles et al., 2011; Martens et al., 2017). The algorithm uses the Priestley-Taylor equation to calculate actual evapotranspiration using net surface radiation and

near-surface air temperature. It calculates interception using the Gash model and uses an adaptation of the Priestley-Taylor equation to calculate evaporation from open water and sublimation from ice and snow. Inputs include air temperature and surface radiation from reanalysis and a gauge-based gridded precipitation product, along with satellite products for SWE, vegetation properties and lightning and soil moisture. There is a version, v3.1b, which uses satellite products for air temperature and surface radiation, but this is only available for latitudes between 50° S and 50° N and was not suitable for this study.

## 2.5  Snow water equivalent

### 2.5.1  GlobSnow

The European Space Agency (ESA) Data User Element (DUE) funded GlobSnow project (Luojus et al., 2013; Takala et al., 2011) provides SWE data for the years 1979–2016. This study uses GlobSnow v2.0. It is derived from a combination of satellite microwave radiometer data (SMMR, SSM/I and SSMIS sensors) and ECMWF weather station observations. A semi-empirical

snow emission model is used to convert passive microwave emissions to SWE. A data assimilation scheme is then applied to the weather station and radiometer-derived SWE observations, to produce a map of the Northern Hemisphere in Equal-Area Scalable Earth Grid (EASE-Grid) format, at nominal 25 km resolution. Pixels are masked in areas of open water, and in areas where there are mountains.

For this study, the Northern Hemisphere data were regridded onto the analysis grid. Since the original data are masked where mountains are present, in the regridded data pixels were masked where at least 50% of the contributing pixels were mountain.

This mostly affects western North America and some regions in Central Asia, masking 16% of the area of the Yenisei, 13% of the Ob, 9% of the Lena and 14% of the Mackenzie. In the Eurasian basins the masked regions are generally in the south where the SWE is low, although in the Mackenzie the mountains extend to the north of the basin. The GlobSnow mountain mask is used for all basin averages of SWE in this study, both modelled and observed, so that comparisons are made over consistent

regions.





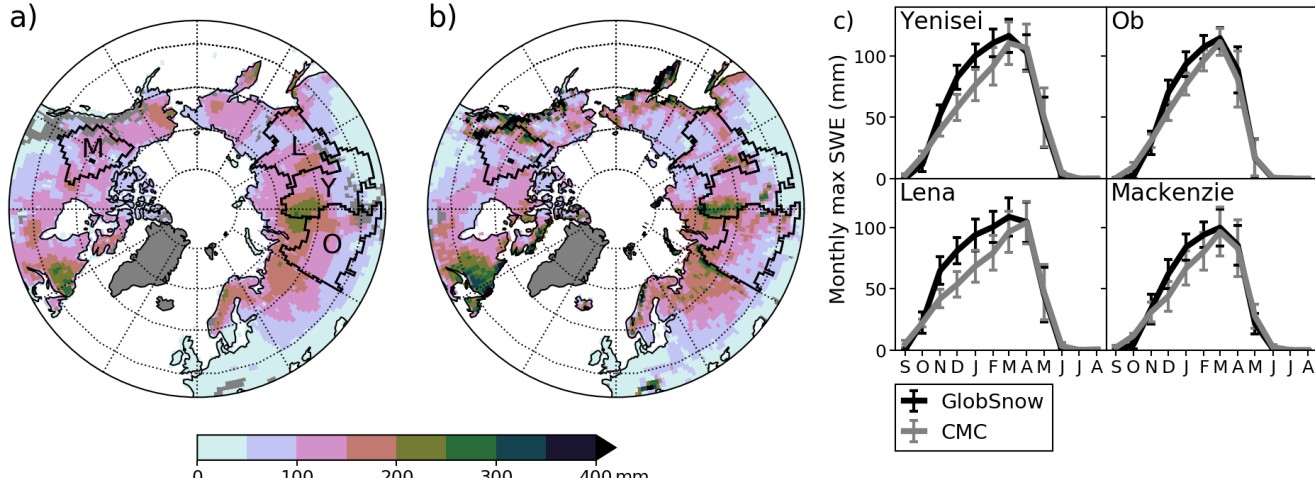

**Figure 1.** Seasonal maximum SWE (mm) for (a) GlobSnow and (b) CMC for the common overlap period, 2002–2010. The basins used in this study are labelled with their initial (*Y*enisei, *O*b, *L*ena, *M*ackenzie). Grey shows regions that are masked. In both data sets Greenland is masked. In addition, GlobSnow pixels are masked where they are more than 50% "mountain". (c) Mean-monthly climatology averaged over the basins (masked using the GlobSnow mountain mask) for the common overlap period, 2002–2010. Error bars show the inter-annual variability, defined as the standard deviation between years.

### 2.5.2 CMC Daily Snow Depth Analysis

The Canadian Meteorological Centre Daily Snow Depth Analysis Data (CMC; Brown and Brasnett, 2010) are an assimilation of a simple snow accumulation and melt model, using temperatures and precipitation from the CMC Global Environmental Multiscale (GEM) forecast model, and daily snow depth data from the World Meteorological Organization (WMO) information
5 system. From snow depth the SWE is estimated using snow density which is dependent on month and climate class. The data are provided as a Northern Hemisphere polar stereographic grid, with a nominal resolution of 24 km and are available for the years 1998–2012.

### 2.5.3 Comparison of SWE products

Figure 1 (a) and (b) show the mean seasonal maximum SWE for GlobSnow and CMC over the common overlap period (2002–
10 2010), with the season defined to be 1st September – 31st August. Many aspects of the large scale distribution are similar in both products and the main maxima are found in similar locations (when not covered by the GlobSnow mountain mask).

Basin average SWE was calculated by averaging over each basin, after having applied the GlobSnow mountain mask. The seasonality of basin average monthly maximum SWE, seen in 1 (c), is slightly different between GlobSnow and CMC, with GlobSnow tending to have a faster accumulation of snow in the earlier part of the winter, while CMC has a more constant



accumulation throughout the winter. However, the seasonal maxima are consistent between the two data sets, although the GlobSnow maximum is slightly earlier in the Lena.

This comparison suggests that the GlobSnow and CMC basin average estimates are remarkably similar for the areas and time periods considered, particularly for estimates of seasonal maxima. A more comprehensive comparison of hemisphere-wide estimates by Mudryk et al. (2015) showed that GlobSnow tended to give larger and earlier seasonal maxima than CMC over the whole Northern Hemisphere, but this does include regions that are not being considered in the current study.

## 3   The JULES land surface model

This study uses the Joint UK Land Environment Simulator vn4.9 (JULES; Best et al., 2011; Clark et al., 2011) which is the land surface component of the Hadley Centre climate model but is here used in offline mode driven by near-surface meteorology. We provide brief details here and refer the reader to the above references for a fuller description.

JULES calculates the exchanges of radiation, heat, water and carbon between the land surface and the atmosphere. As employed here each grid box contains a mixture of nine surface types (five vegetation types and four non-vegetated surfaces), for each of which the surface fluxes are calculated independently. Subsurface fluxes of moisture and heat are calculated using a layered soil model, with a single soil column in each grid box. Supersaturation of soil moisture is avoided by moving the extra water to lower soil layers. The snow pack is modelled using a multi-layer snow scheme that employs a variable number of layers depending on the depth of snow. Each snow layer has a prognostic temperature, density, grain size, and solid and liquid water contents. Snowfall in vegetated areas with needle-leaf tree cover can be partitioned between the vegetation canopy and the underlying ground (Essery et al., 2003; Essery and Clark, 2003). The intercepted snow leaves the canopy though sublimation and wind-speed-dependent unloading of melting snow. The surface albedo is affected by the evolving grain size of the snow surface and the extent to which the vegetation is buried by snow. In this study we employ a TOPMODEL-based parameterisation of surface runoff (Beven and Kirkby, 1979). River routing is carried out as a post-processing step, using Total Runoff Integrating Pathways on a 1° river routing grid (TRIP; Oki and Sud, 1998).

The ability of JULES to represent snow and other high-latitude phenomena has been evaluated in several studies including those of Burke et al. (2013), Hancock et al. (2014), Chadburn et al. (2015), Ekici et al. (2015) and Ménard et al. (2015). In particular in SnowMIP2 JULES was found to be one of the best models for simulations of open sites, albeit it was thus not as skilful at forest sites (Rutter et al., 2009).

For this study, JULES is run at the native resolution of each driving data set (0.5°, except when using the PGF data which is 1.0° resolution). Meteorological data are provided every 6 hours (CRUNCEP) or 3 hours (all other data) and JULES interpolates these onto a 30 minute time step. The precipitation rate is kept constant over the data time step (3 or 6 hours). The model output is regridded to 1.0° resolution for analysis.

The WFDEI-CRU and WFDEI-GPCC data sets contain separate rainfall and snowfall fields that can be used directly by JULES, but the CRUNCEP and PGF data sets only provide total precipitation. For these, JULES assumes that the precipitation is snowfall when the near-surface air temperature is less than or equal to 274 K, while at higher temperatures the precipitation



is assumed to be entirely rainfall. Hancock et al. (2014) suggest that, as long as the start and end dates of accumulation are correct, the modelled SWE is relatively insensitive to the choice of threshold.

The distribution of soil properties and land types was based on data from the IGBP Global Soil Task Force (Global Soil Data Task, 2000), gridded to the resolution of each driving data set. Vertical fluxes of soil water follow Darcy's law, using

hydraulic characteristics calculated after Brooks and Corey (1964). Runoff was generated using the TOPMODEL formulation, with topographic index regridded from the high-resolution HydroSHEDS data (Marthews et al., 2015). The model was spun up using the first five years of each data set four times, for a total of 20 years, then run for several decades starting in 1970 (CRUNCEP and PGF) or 1979 (both WFDEI variants) and running to the end of the available data.

### 3.1 Experiments

Three experiments using JULES were performed:

**CTL** Control runs, driven with the original meteorological data sets.

**GRC** Runs in which the total cold season precipitation is scaled to match that derived from GRACE, as described in Sect. 4.1 below. The scale factors, which vary by basin and by run, can be seen in Table 2. These factors are applied constantly to both rain and snow for the months October to February inclusive. In the rest of the year the precipitation is unchanged.

**UCC** Runs for which wind-based undercatch correction (Adam and Lettenmaier, 2003) is applied to the precipitation (CRUN-CEP and PGF only). This uses the same catch ratios as were used in the creation of WFDEI (Weedon et al., 2011), one for each month for rainfall and snowfall separately. The correction is applied throughout the whole year. The ratios are available at $0.5°$ resolution, so were re-gridded to $1°$ resolution for PGF.

The full set up for each run is available as a version-controlled Rose suite, through the Met Office repository: https://code.

metoffice.gov.uk/trac/roses-u. Full details of the suite number and revision for each run is given in Table S2.

### 3.2 Mean bias error

To evaluate the runs, we calculate the mean bias error (MBE) of the annual maximum SWE in each basin ($M_b$) to be

$$M_b = \frac{1}{n_y} \sum_{i=1}^{n_y} S_{i,b} - s_{i,b} \tag{1}$$

where $S_{b,i}$ is the modelled SWE in basin $b$ at timestep $i$, $s_{b,i}$ is the corresponding observed SWE and $n_y$ is the number of years

of overlap between model and observations. To average over basins, it is weighted by the basin area, $A_b$

$$M_S = \frac{\sum_{b=1}^{n_b} A_b M_b}{\sum_{b=1}^{n_b} A_b} \tag{2}$$

The variance of the bias error in each basin is

$$\sigma_b^2 = \frac{1}{n_y - 1} \sum_{i=1}^{ny} (S_{i,b} - s_{i,b})^2 \tag{3}$$





and the combined variance is

$$\sigma_S^2 = \frac{(n-1)\sum_{b=1}^{n_b} A_b \sigma_b^2}{\left(\sum_{b=1}^{n_b} A_b\right) - 1}. \tag{4}$$

## 4 Precipitation corrections

### 4.1 GRACE-derived precipitation estimates

A water balance approach (Swenson, 2010) was used to estimate the precipitation ($P$) at a basin scale:

$$P = \frac{dS_{\text{tot}}}{dt} - E - Q_{\text{net}} \tag{5}$$

where $dS_{\text{tot}}/dt$ is the change in TWS ($S_{\text{tot}}$), $E$ is the total evaporation flux, including sublimation, and $Q_{\text{net}}$ is net runoff. At the annual or monthly scale, this can be calculated to be

$$\int_{t_1}^{t_2} P(t)dt = S_{\text{tot}}(t_2) - S_{\text{tot}}(t_1) - \int_{t_1}^{t_2} \left(E(t) + Q_{\text{net}}\right)dt, \tag{6}$$

where $t_1$ and $t_2$ are the start and end of each accumulation period respectively. TWS encompasses all water storage, including soil moisture, ground water, water in wetlands, lakes and rivers, and water stored as snow in the snow pack. Evaporation includes transpiration, evaporation from soil surfaces, evaporation from intercepted water and other open water (rivers and lakes), as well as sublimation from frozen surfaces. As the water balance is calculated over whole basins there is no incoming runoff and the net runoff is equal to the basin discharge.

The change in TWS is calculated by differencing the GRACE anomalies between months, and averaging over the basin. Monthly evaporation is provided by GLEAM and averaged over the basin, while the basin discharge is obtained from the GRDC measurement station closest to the basin outflow.

    The water balance is calculated for each month in the cold season (defined as October to February inclusive for all basins). During this season snowfall is the dominant precipitation type, the accumulating snow-pack is the dominant change to TWS,

and the runoff and evaporation fluxes are relatively small (Fig. S4). This minimises the effect of uncertainty in evaporation and basin discharge products, which could be more significant outside of the cold season (Swenson, 2010; Seo et al., 2010).

### 4.2 Comparison of precipitation estimates

Figure 2 shows the long-term mean monthly accumulated total precipitation during the cold season (October–February) for each of the four driving data sets and for the GRACE-derived precipitation estimates (all averaged over the common overlap

period). The differences between data sets vary between the catchments, but in general the driving data sets accumulate less precipitation than the GRACE estimate. The CRUNCEP and PGF data sets have lower precipitation than the WFDEI data through the whole cold season in all basins, but are very similar to each other as they were both bias corrected to CRU data without undercatch correction. The WFDEI data sets generally have higher precipitation, and are much closer to the GRACE-derived estimate, even indicating a larger accumulation by December in the Lena and the Ob. The final accumulation is similar





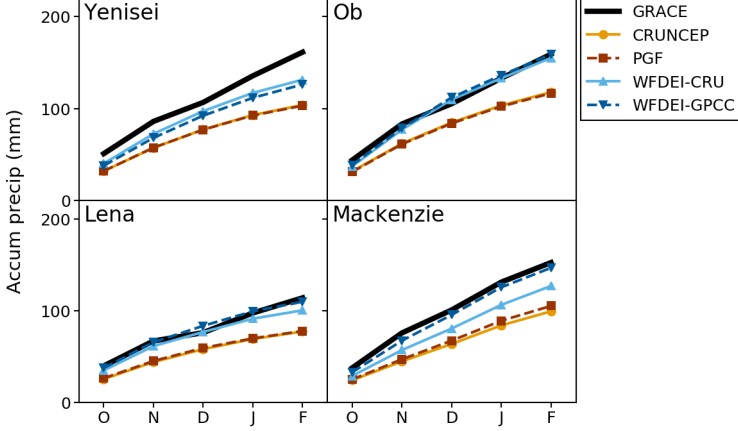

**Figure 2.** Mean monthly accumulated cold season precipitation in October–February inclusive, averaged over the common overlap period, 2002–2010, as derived from GRACE TWS (Sect. 4) and from the driving data sets (Sect. 2.1).

to the GRACE-derived estimate in three basins for WFDEI-GPCC and two basins for WFDEI-CRU . Overall the implication is that the driving data sets often do not contain enough precipitation to close the water budget in these basins in the cold season, particularly for the CRUNCEP and PGF data.

### 4.3 Calculation of precipitation scale factors

5  In order to appropriately increase the amount of precipitation in the driving data sets, a scale factor was calculated for each data set in each study basin. The scale factor between GRACE and each driving data set was found by calculating the ratio of cold season accumulated precipitation derived from GRACE to that in the driving data. As the driving data cover a longer time period than GRACE, to ensure that the difference is not due to inter-annual variability or long-term climate trends each ratio was calculated using only data from the overlapping period of the relevant data set with GRACE. This period starts in 2002 for
10  all of the data sets, and runs to the end of each data set. The scale factors for each basin can be seen in Table 2.

The relative scaling between basins is consistent for each data set, with the Yenisei requiring the largest correction (between 24% and 55% increase), and the Ob requiring the least (between a 2% decrease and 37% increase). There is a striking difference between the WFDEI data sets, which require increases up to 27% (and one decrease), and the PGF and CRUNCEP data sets which require much larger increases of between 35% and 55%.

15  The GRC runs of JULES were carried out by scaling both snowfall and rainfall by these factors during the cold season only for all years of the run (not just the GRACE period). Snow and rain that falls outside of this season were unchanged. This design ensured that each run received the same amount of cold season precipitation as indicated by GRACE, when averaged over the basin and over the period of overlap between the driving data set and GRACE; the temporal and spatial variability of precipitation still varied between runs.




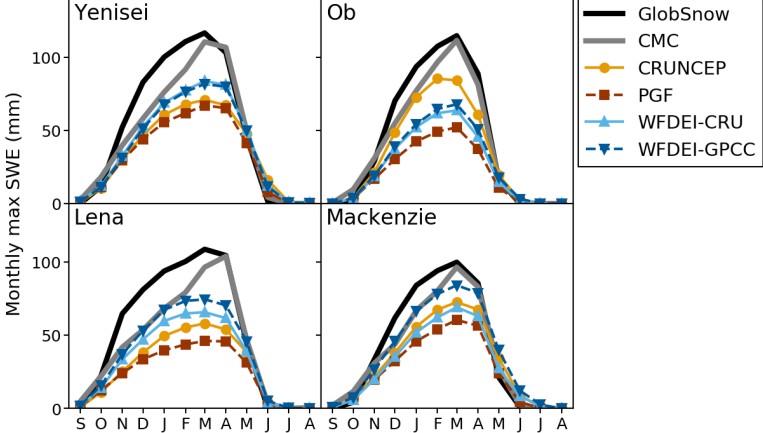

**Figure 3.** Climatology of monthly maximum SWE (mm) from observations and from the CTL model runs averaged over the common overlap period, 2002–2010.

## 5 JULES model runs

### 5.1 Control runs (CTL)

The simulated monthly maximum SWE for each basin can be seen in Fig. 3. In many cases JULES accumulates snow more slowly than the observational estimates, particularly later in the accumulation season, and the SWE is less sharply peaked than the observations.

Figure 4 shows the MBE for seasonal maximum SWE averaged over all four basins with respect to both GlobSnow and CMC. In general the magnitude of the MBE is largest for PGF, and smallest for WFDEI-GPCC (Fig. S5 shows the MBE for each basin). The largest deficit is for the PGF data in the Lena basin, for which it is -57% for GlobSnow and -56% for CMC. The smallest deficit is found in the Mackenzie for WFDEI-GPCC driving data, which has an MBE of -17% against both observations. Maps of the modelled SWE show broadly similar distributions of snow to the observations (Fig. S6). The difference between the modelled and observed SWE (eg. GlobSnow in Fig. S7) show that, although there are a few regions of higher SWE in JULES outside of the studied basins, there is a deficit of modelled lying snow across most of the Northern Hemisphere. The general tendency for JULES to underestimate SWE is consistent with the indication of insufficient cold season precipitation (Fig. 2), and both analyses also agree that the WFDEI runs are generally closer to the observations. However it is clear that, even in the cases where the precipitation is close to the GRACE-derived estimate, the modelled maximum SWE is still lower than observed.

The annual discharge in JULES is also severely underestimated compared with GRDC observations, as seen in Fig. 5 which shows the basin discharge averaged over all four basins. All of the runs demonstrate a substantial dry bias in comparison with GRDC values. CRUNCEP is particularly poor, with an 80–90% deficit in annual mean basin discharge, while the other runs underestimate basin discharge by between 52–57% on average. This difference between CRUNCEP and the other data sets





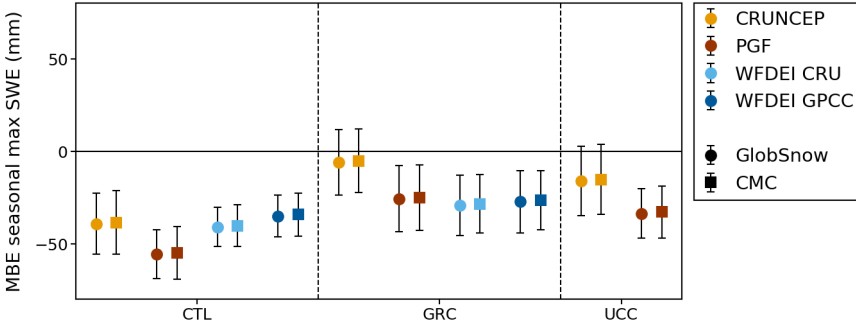

**Figure 4.** Mean bias error (MBE) of seasonal maximum SWE (mm), averaged over all basins (Eq. 2). Calculated over the common overlap period, 2002–2010. The error bars show the basin averaged standard deviation, $\sigma_S$ (Eq. 4).

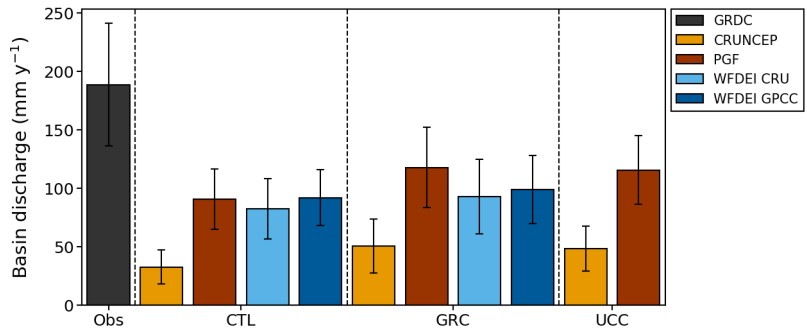

**Figure 5.** Mean annual observed and modelled basin discharge (mm y$^{-1}$) for the common overlap period, 2002–2010.

is consistent across all the basins. The deficit is particularly pronounced during the spring peak (Fig. S9) which in nature is strongly driven by snow melt. In JULES, the peak in the modelled river flow is broader and occurs later than the observations, which may indicate deficiencies in the modelled hydrology as well as a lack of melting snow. However, at the annual scale the bias may also be affected by warm-season rainfall.

## 5.2 Runs with scaled precipitation (GRC)

Using the GRACE-scaled precipitation to drive JULES gives improved SWE representation overall, with significant increases in monthly and seasonal maximum SWE (Fig. 6). The modelled seasonal cycle is much more similar to observations, particularly in the Yenisei and Mackenzie, and early in the season for the Lena and Ob. In the Ob, the CRUNCEP GRC run is much more similar to the observations than any of the other models, while in the Mackenzie it now overestimates SWE. In the Lena, the seasonal peak is still not attained with any data set. Overall the largest increases in SWE are found for the CRUNCEP and PGF runs, consistent with the larger scale factors used (Table 2). The annual maximum SWE is increased by 30% averaged across all basins and runs, with the CRUNCEP and PGF runs having an average increase of 48% and the WFDEI runs having



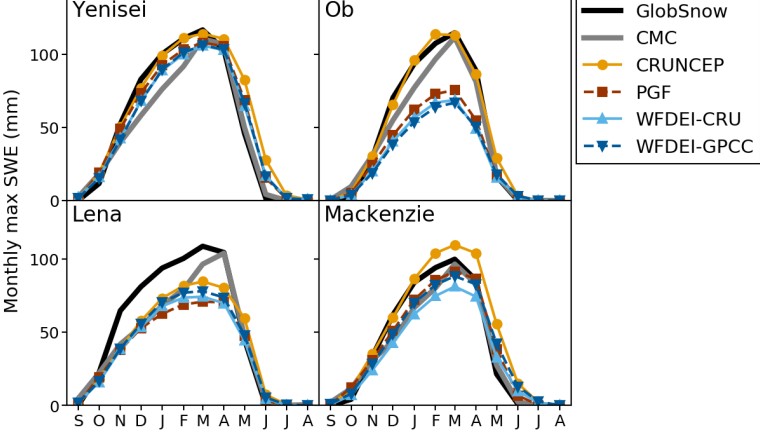

**Figure 6.** Climatology of monthly maximum SWE (mm) from observations and from the GRC model runs for the common overlap period, 2002–2010.

a smaller average increase of 12%. Overall, the magnitude of the MBE is decreased for all driving data sets (Fig. 4). Although there is an increase in SWE over the extent of each basin (Fig. S8), the effect on MBE varies between driving data sets and basins: the widespread underestimation of the CTL runs is replaced by a more nuanced picture, including some cases with modest overestimation of SWE (Fig. S5). The modelled estimates of maximum SWE in the Yenisei and Mackenzie are now

clustered around the observational estimates, while a low bias is still found in most runs for the Lena and Ob. The biggest improvement is seen in CRUNCEP runs, which now only has an MBE of -5% overall.

Basin discharge is higher in the scaled runs than the control runs, although the increase is relatively small (27% overall) compared to the general dry bias. The improvement in representation of basin discharge varies significantly between data sets, from 8% in WFDEI-GPCC to 57% in CRUNCEP, which still has the lowest basin discharge (the highest absolute increase is

27 mm/yr in PGF, while the lowest is 7 mm/yr in WFDEI-GPCC). Although the basin discharge increases overall, the increase in SWE and the resultant increase in spring snow melt do not change the timing of the peak basin discharge in the models. The modelled estimates of basin discharge are still considerably lower than the GRDC estimates, with CRUNCEP underestimating by 74% and the others between 39–52% (Fig. 5). This implies that factors other than cold season precipitation and snow accumulation are contributing to the lack of river flow. In particular the low bias and late peak in discharge for the Yenisei and

Mackenzie persists (Fig. S9) despite the peak SWE now being close to the observational estimates.

## 5.3   Runs with undercatch correction (UCC)

As discussed above, a possible explanation for for underestimation of cold season precipitation in observation-based data sets is undercatch. In an attempt to account for this effect two of the data sets — WFDEI-CRU and WFDEI-GPCC — already implement undercatch correction. The two data sets that do not apply a correction required the largest scaling factors to

match the GRACE-derived estimates. To investigate the importance of the undercatch correction in comparison to the GRACE





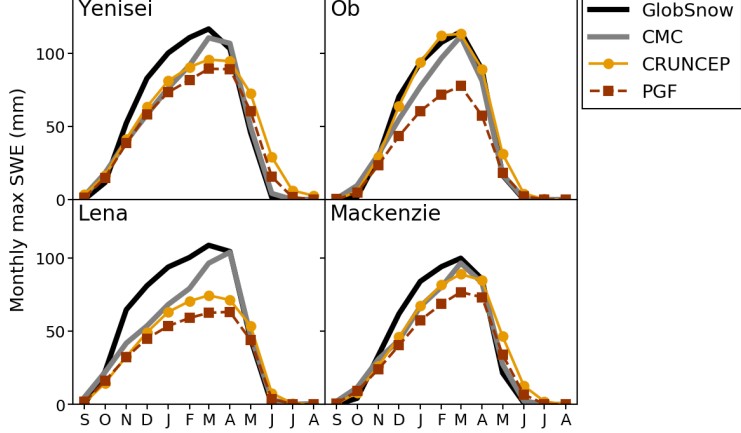

**Figure 7.** Climatology of monthly maximum SWE (mm) from observations and from the UCC model runs for the common overlap period, 2002–2010.

corrections, the catch ratios calculated by Adam and Lettenmaier (2003) were used to correct the CRUNCEP and PGF data on a monthly basis. Both rainfall and snowfall were adjusted, using separate catch ratios, for all months of the year, and used to force JULES in the UCC runs.

The undercatch correction increases the precipitation across the whole year, but here we focus on the cold season precipi-
tation increase. The increase is large, but less than indicated by the GRACE precipitation scale factors. In the Ob, undercatch correction increases the amount of cold season precipitation by 26% over the whole basin, compared to an overall increase of 35% in the GRC runs for CRUNCEP and PGF. However, in the other basins the increase in precipitation due to undercatch correction is between 20–25% and the increase based on GRACE precipitation is between 45–55%. Thus, the undercatch correction accounts for between 30–75% of the cold season deficit that is indicated by the comparison with the GRACE-derived
precipitation estimates.

The improvement to the estimate of SWE is clear, although it is not as large as in the GRC runs (Fig. 4). On average the two undercatch corrected driving data sets used for the UCC runs have maximum SWE increased by 34%, compared to an increase of 48% when the same two data sets were scaled by the GRACE scale factors. In contrast, the increases in annual basin discharge were much more similar between the GRC and UCC runs. In the Ob, the undercatch correction even resulted in
higher basin discharge than the GRACE scaling, despite the GRACE scaling resulting in higher total annual precipitation; this is because in the UCC runs the precipitation is increased across the whole year, contributing to increased river flow in warm months as well as the spring snow melt, whereas the GRC runs only have increased precipitation in the cold season.

Along with the fact that GRACE indicates that the WFDEI runs also under-represent precipitation in at least some of these basins, this suggests that although undercatch correction can account for some of the deficit in precipitation, it cannot account
for all of it.





## 6 Discussion

This study has shown that gauge- or reanalysis-based estimates of cold season precipitation in boreal basins can be significantly lower than is suggested by a water-balance approach using GRACE TWS data. By scaling the driving meteorological data sets, we can significantly improve the representation of lying snow in a land surface model. The deficits seen in precipitation are large but similar in magnitude to those calculated by Finney et al. (2012) who calculated precipitation corrections of between 36 and 70%, based on a comparison of modelled and observed SWE, and by Behrangi et al. (2017) who used GRACE to evaluate precipitation data sets in endorheic cold mountainous basins, and found that some precipitation data sets only captured between 10 and 60 % of the GRACE-based precipitation estimates. In this study we used time-invariant corrections, but this could be extended to allow time-varying corrections, say on an annual basis.

The GRACE-based estimates provide a means to account for measurement errors in the gauge data that are used to bias-correct reanalyses. This study suggests that undercatch is an important source of error, but in many cases the application of an undercatch correction does not remove the bias entirely. Further sources of error in the gauge-based estimates include spatial variability that is missed by the gauge network. While the GRACE TWS method used here circumvents many of these limitations it introduces uncertainties from other terms in the water budget (e.g. evaporation) and is inherently large scale — it may not result in improved SWE when looking at the local scale.

Undercatch correction, on the other hand, can be calculated at a local or grid box scale and can, in theory at least, more easily take account of changing meteorological conditions down to the time scale of individual storms. Furthermore it is easy to implement and is attractive as it addresses known deficiencies in the observations. It can also be used to correct historical data sets that pre-date GRACE. However the correction it is uncertain, including dependency on the type of gauge used in each area, and again requires further inputs, each with uncertainties. A further approach, not studied here, is to use estimates of SWE, either from ground measurements or remote sensing, to estimate snowfall.

This study of cold season processes shows that the undercatch correction is equivalent to a substantial fraction of the GRACE-derived correction, suggesting that for gauge-based data sets the undercatch correction can be considered a minimum requirement that should be applied whenever possible. However, this varies by region, implying that there are different reasons for the deficit in precipitation in different regions. The CRUNCEP and PGF data sets do not include an undercatch correction but have been widely used with land surface models; our results suggest that any aspects of those studies that are potentially sensitive to snowfall, such as high-latitude hydrological analyses, should be regarded as particularly uncertain. Undercatch errors potentially also affect summertime precipitation, but have not been studied here.

Figure 8 shows that the increase in seasonal maximum SWE is proportional to the increase in seasonal snowfall between experiments, with a very strong correlation ($r^2 = 0.98$) between the two. The gradient of 0.76 implies that not all of the increase in snowfall manifests as an increase in SWE, which is mainly due to increased sublimation and a small increase in snow melt, but it does confirm that correcting snowfall is a direct and reliable approach with which to target errors in simulated SWE. This correlation also implies that the approach of Finney et al. (2012) to increase driving snowfall based on the required increase in SWE is reasonable.





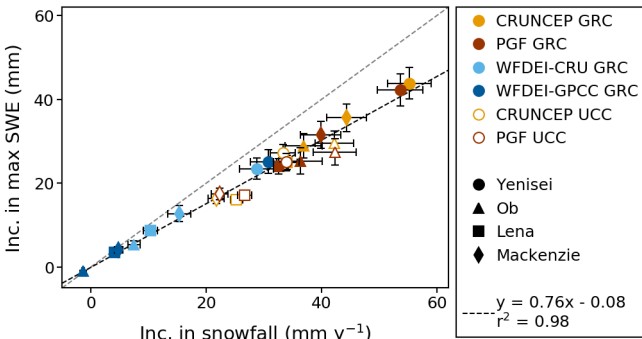

**Figure 8.** Increase in annual maximum SWE (mm) from CTL to GRC and UCC runs compared to increase in annual snowfall (mm $y^{-1}$), averaged over all basins (using the GlobSnow mountain mask for snowfall as well as SWE) for each driving data set. This is averaged over the whole of each JULES run (see Table 1).

The use of corrected precipitation improved results in most catchments and with most driving data sets, often substantially. In contrast the larger dry bias in modelled basin discharge was found in all simulations, with only modest improvements from corrected precipitation. This difference is a result of the much closer links between point snowfall and SWE, whereas modelled discharge can be viewed as the net effect of many hydrological processes acting across the catchment. A striking illustration

of this is the large difference between the CRUNCEP and PGF river discharge, despite very similar precipitation inputs (both having been bias-corrected to CRU precipitation). The larger river flow in PGF is balanced by a smaller evapotranspiration flux (Fig. S10), while evapotranspiration in CRUNCEP tends to be closer to the WFDEI-based estimates. In turn this can be related to differences in other aspects of the meteorological forcing (Figs. S1 and S2); in particular, PGF has much higher specific humidity, particularly in the summer, than the other data sets. In spring and summer PGF humidity is 18% – 87% higher

than in CRUNCEP, which suppresses evapotranspiration, which is 10% – 17% lower in PGF for the CTL runs. (This can be contrasted with the air temperature, which is nearly identical in PGF and CRUNCEP, implying that the data sets employed different methods to reconstruct humidity.) This is consistent with the fact that evapotranspiration in the PGF runs is generally less than estimated by GLEAM, while the other runs tend to be closer to GLEAM, although there is considerable uncertainty in the GLEAM product. Although the focus of this study is on correction of precipitation products, which is shown to improve

modelled SWE, it is clear that other meteorological variables are also important, particularly for other aspects of modelled hydrology.

The modelled runoff ratio ($Q_{net}/P$) is much lower on average for CRUNCEP than for runs with the other data sets. When combined with the already relatively low precipitation in the CTL run, this means the CRUNCEP data set generates particularly small values of discharge in all basins — overall the CRUNCEP basin discharge is 16% of observed in the CTL run and 25% of

observed in the GRC and UCC runs. The extra precipitation input in the GRC runs largely appears as increased evaporation in CRUNCEP runs, whereas it is largely converted to runoff for the other data sets (Fig. S10). The low runoff ratio for CRUNCEP





is possibly related to the longer time step of the data set (6 hours rather than 3 hours), within which the precipitation rate is constant in the JULES runs; the lower intensity of precipitation will tend to result in larger interception loss.

The seasonal maximum SWE tends to be biased low even when precipitation inputs are corrected. Assuming that both the precipitation and SWE observations are correct, a possible implication is that the modelled sublimation rate is too high. Sublimation in the CTL runs, averaged over all basins, ranged from 37% of snowfall (PGF) to 44% (CRUNCEP). It is very difficult to measure sublimation, particularly over large areas, and most estimates tend to be based on water balance methods or models (Liston and Sturm, 2004). Estimates at a range of scales indicate that 10 – 50% of snowfall can be lost through sublimation, with substantial variation depending on land cover and meteorological conditions (e.g. Liston and Sturm, 2004; Brun et al., 2013; Casson et al., 2018). For the Mackenzie basin sublimation was estimated as 29 mm (7% of annual precipitation; Déry and Yau, 2002), while numerical modelling by Yang et al. (2010) suggested that approximately 24% of annual snowfall in the area 50 – 70° N was lost as sublimation. Thus the evidence suggests that sublimation might be rather high in JULES, albeit the evidence base is itself rather uncertain. Further investigation, including point-scale runs in data-rich environments to examine the simulated water budget across a range of land covers such as in SnowMIP2 (Rutter et al., 2009), will increase our confidence in the ability of JULES and other models to correctly simulate sublimation and other key processes.

However, errors in modelled sublimation and indeed all cold season processes are insufficient to fully account for the dry bias in annual discharge. It is also possible that rainfall outside of the cold season is underestimated in the driving data. In all of these basins there is more precipitation in summer (when it is more likely to be rain) than in winter, suggesting that correction of warm season rainfall could potentially have a larger impact on the annual water balance. However, there are clear signs that meteorological inputs are not the only source of error, and that there are fundamental deficiencies in the model's representation of runoff generation processes: even a good estimate of peak SWE does not result in a good representation of the spring discharge peak (Fig. S9). It is likely that the parameterisation of infiltration into partly-frozen ground and related runoff generation processes are not well represented in JULES. Previous work has shown that alternative descriptions of frozen soil can improve the modelled runoff peak (Finney et al., 2012).

## 7 Conclusions

There is a substantial body of literature on the intercomparison of global precipitation data sets, with a lesser focus on the particular issues found at high latitudes where much of the precipitation falls as snow. There is an ongoing need to compare these precipitation products and to ensure that the best meteorological data are made available as inputs to land surface modelling. This study has focussed on precipitation but the model results clearly indicate that other variables, such as humidity, are also important.

Land surface modellers should continue to critically evaluate the meteorological data they use, and ideally run a model using a variety of data sets. The extent to which results are sensitive to the choice of meteorological data will vary; for some analyses there might be relatively little sensitivity, but by and large this can only be determined through the use of an ensemble of runs forced by alternative data. Meteorological data should also be evaluated in combination with products that describe related



parts of the hydrological system, such as GRACE TWS, estimates of SWE based on remote sensing, and river discharge as used in this study. Although each product comes with its own uncertainty, and a range of alternative and potentially conflicting data sets is often available, the combination of estimates across different parts of the hydrological system can provide extra insights, particularly if a model shows consistent biases across several components.

This study shows that at a basin scale the cold season precipitation in four data sets that are commonly used to drive land surface models is low compared with estimates derived from GRACE TWS. This leads to consistent and large errors in the SWE and basin discharge calculated by JULES, which are also low compared to observations. By increasing the precipitation in JULES to match the GRACE estimates the modelled SWE is substantially improved, although river discharge is still low — likely because of a combination of poor modelling of runoff processes during the spring melt and possible underestimation of

summertime precipitation. By providing methods to better define the precipitation inputs to the system, the current study paves the way for subsequent work on key hydrological processes.

*Code availability.* The model runs in this manuscript were carried out using JULES version 4.9, with modifications to allow scaling of the precipitation input. This is available through the JULES repository and is located at https://code.metoffice.gov.uk/trac/jules/browser/main/branches/test/emmarobinson/vn4.9_arctic_scaling?rev=9936 (registration required). The runs were carried out using Rose (http://metomi.

github.io/rose), the control files are available through the Met Office rose repository (https://code.metoffice.gov.uk/trac/roses-u), and details are in Table S2. All other analysis code, including river routing and post-processing, available on request.

*Author contributions.* DBC designed the study. ELR carried out model runs and analysis. Both wrote the manuscript.

*Competing interests.* The authors declare no competing interests.

*Acknowledgements.* This work was supported by the Natural Environment Research Council [grant numbers NE/S017380/1, NE/H000224/1].

GRACE land data are available at http://grace.jpl.nasa.gov, supported by the NASA MEaSUREs Program.





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





**Table 1.** Summary of meteorological driving data.

| Driving data | Years avail. (used) | Temporal resolution | Spatial resolution | Reanalysis | Source of precipitation | Precip. variables | Undercatch correction |
|---|---|---|---|---|---|---|---|
| CRUNCEP v4 (Viovy, 2014) | 1901–2010 (1970–2010) | 6 hour | 0.5° | NCEP/NCAR Reanalysis Project 1 | CRU TS 3.21 | Total | No |
| PGF v2 (Sheffield et al., 2006) | 1948–2012 (1970–2012) | 3 hour | 1° | NCEP/NCAR Reanalysis Project 1 | CRU TS | Total | No |
| WFDEI-CRU (Weedon et al., 2014) | 1979–2016 | 3 hour | 0.5° | ERA-Interim | CRU TS 3.101 and 3.21 | Rain, snow | Yes |
| WFDEI-GPCC (Weedon et al., 2014) | 1979–2013 | 3 hour | 0.5° | ERA-Interim | GPCC v5 and 6 | Rain, snow | Yes |

**Table 2.** Precipitation scale factors, calculated as the ratio of GRACE-derived to data set precipitation, using all years of overlap between each driving data set and GRACE.

| Driving data | Yenisei | Ob | Lena | Mackenzie | Mean |
|---|---|---|---|---|---|
| CRUNCEP | 1.55 | 1.35 | 1.47 | 1.53 | 1.47 |
| PGF | 1.55 | 1.37 | 1.55 | 1.54 | 1.49 |
| WFDEI-CRU | 1.24 | 1.06 | 1.11 | 1.16 | 1.14 |
| WFDEI-GPCC | 1.27 | 0.98 | 1.04 | 1.05 | 1.08 |