# Peer review of "Using GRACE to derive corrections to precipitation data sets and improve modelled snow mass at high latitudes"

_Hydrology and Earth System Sciences, 2019_

## Referee Comment (RC1) · Anonymous Referee #1 · 8 Jun 2019

***Interactive comment*** on "Using GRACE to derive corrections to precipitation data sets and improve modelled snow mass at high latitudes" by Emma L. Robinson and Douglas B. Clark

This study compared four commonly-used meteorological data sets with GRACE corrected precipitation over four basins at high latitudes and found that four commonly-used data sets generally underestimated the cold season precipitation. Undercatch correction was further employed to alleviate some of the underestimation of cold season precipitation. Driven by these original/corrected precipitation inputs, JULES land surface model was used over four high latitudes basins. Improvements on the simulation of seasonal maximum SWE and river discharge were found using the corrected precipitation data.

Finding ways to better define precipitation inputs is important for earth systems interpretation in terms of hydrological and biogeochemical cycles. This paper provides us with promising ways to improve the cold season precipitation data. However, such work is challengeable and full of uncertainties (e.g., uncertainties in other meteorology data, model physics...). I agree that authors propose to conduct systematic evaluation in combinations with other hydrological related products(e.g., GRACE TWS, SWE derived from remote sensing, river discharge...). While it will further result in multi-source of uncertainties. My concerns are thus as follows: i) is there a need to conduct the preliminary water balance check for assessing the various products; ii) if the model show consistent biases across several components, can we firmly conclude that data sets is of uncertainty? What is the role of model uncertainty? iii) what is the role of energy related regimes, especially in seasonally cold regions. As far as I know, thermal state can also largely affect the hydrologic variables(runoff, infiltration...).

The specific comments are as follows.

Abstract

Line 15: '... that that ...' change into '... that ...'
Line 17: '... much more modest improvements are found in modelled river discharge.' Here you emphasis the comparison in the model performance of maximum SWE and river discharge, why?

1 Introduction

Page 2, Line 1-Line 7: considering merging these two paragraphs?
Page2, Line 8: considering rephrase?
Page 2, Line 18: 'SWE (e.g. GlobSnow; Takala et al., 2011) but the need for an algorithm...'
      Add the comma between as 'SWE (e.g. GlobSnow; Takala et al., 2011), but the need for an algorithm...'
Page 2, Line 22-24: considering rephrase
Page 2, Line 27: what does 'the impact' mean?
Page 3, Line 2: 'that that' change into 'that'
Page 3, Line 24: 'of $45°$ N Yang et al. (2005)'
      add a comma as 'of $45°$ N, Yang et al. (2005)'
page 3, Line 26-28: considering rephrase
page 3, Line 30-31: considering rephrase
page 4, Line 10: delete 'Behrangi et al. (2017)' or put it at the end of sentence?

Page 4, Line 17: 'and that this leads to…' feel awkward.

Page 4, line 19-22: the result part should not be presented here

Page 8, Line 13: '1(c)' change into 'Fig. 1 (c)'

Page 9, Line 32- page 9, Line 1: is there a transition state between snowfall and rainfall, ie mixture of snow and rain? How to consider such state in a physical way?

Page 10, equation1:

$S_{b,i}$ is not shown in equation ($S_{i,b}$), is the unit necessary here?

Equation 2: what is the meaning of Ms?

Page 11, Equation 5: is the unit necessary here?

Page 13, Line 4-5: '…, and the SWE is less sharply peaked than the observations.' Why?

Page 13 ,Line 8-9: considering rephrase

Page 15, Line 7-8: better to add the corresponding figure here.

Page 15, Line 17: 'for for' change into 'for'

Page 17, Line 13-15: not clear, considering rephrase

Page 17, Line 28: how did you get such a conclusion? Please provide corresponding figure or reference.

Page 19, Line 1-2: do you have the simulations of interception loss?

Discussion: why did you only select the maximum of SWE as the model evaluation output? How about the starting date of snow accumulation?

What is the effect of the water storage before the accumulation of snow on the SWE simulations?

Figures:

Figure 1: add the x-axis title of fig. 1c.

Figure 2: add the x-axis title. Why did you define the cold season as 'october-february'? as I can see that from Figure 3, the maximum of SWE usually happens in March/April.

Figure 3: please add the x-axis title

Figure 6 & 7: add the x-axis title

Figure 8: x-axis title, unit seems to be truncated should be (mm y$^{-1}$), not (mm v$^{-1}$).

Supplement:

Figure S1 & S2 & S9: add the x-axis title

Figure S4: '(GLEAM…' the other bracket is missing?

---

## Referee Comment (RC2) · Anonymous Referee #2 · 25 Aug 2019

This work demonstrated systematically underestimations of high latitude precipitation existing in four popular atmospheric forcings. After precipitation correction by GRACE total water storage (TWS) data, these four atmospheric forcings were used to drive the land surface model JULES in order to assess improvements in simulating hydrological cycles in four Arctic river basins. By comparing to independent datasets, the authors showed that the simulated monthly max snow water equivalent (SWE) was significantly improved. Although river discharges were still underestimated particularly during the spring, the limitation of the model and other possible reasons were explicitly discussed.

This study is well designed. Although there are a lot of uncertainties in estimating

precipitation using GRACE TWS (e.g., uncertainty of ET data, groundwater depletion, etc), the authors ingeniously avoided most of possible perturbations by carefully selecting the research region and season. This work will make hydrologists and modellers re-consider the precipitation in meteorological products, which is the first-order driver of land hydrological cycles. The contents fits the scope of the journal. The manuscript deserves consideration after several questions are addressed.

Page 4, Line 31-32: The resolution of the river basins is coarse. Is there significant difference of basin area between this one and other fine resolution products?

Page 6, Section 2.2: Is GRACE RL05 a mascons solution, which is more advanced than the standard spherical harmonic approach especially for high latitudes?

Page 7, Section 2.4: GLEAM ET is a model output. It is also driven by atmospheric forcing. The aim of using GLEAM ET is to show negligible ET during the cold season. Is there any other ET datasets available for a parallel comparison? Is it better to assume zero ET in cold season than introduce GLEAM ET in Equation 6?

Page 11, Equation 5-6: I think it should be P = dS_tot/dt + E + Q_out as dS_tot/dt = P - E - Q_out.

Page 12 Section 4.3: Here you only re-scaled precipitation in the cold season (not during the whole year). Perhaps this leads to discharge underestimation in spring (Figure S9). Does it make sense to launch a simple simulation with re-scaled precipitation over the whole year? Maybe it can explain the underestimated discharges.

Page 17 Line 19: remove "it".
* * *

---

## Author Comment (AC1) · 4 Oct 2019

**Response to Referee 1:**
**Using GRACE to derive corrections to precipitation data sets and improve modelled snow mass at high latitudes**

Emma L. Robinson          Douglas B. Clark

We thank the reviewer for their comments and suggestions. In response to the main concerns:

i) In the course of the analysis, we have indeed investigated the overall water balance (eg. we have included the GRDC basin discharge and GLEAM evapotranspiration in Fig S10 in the Supplementary Information (labelled "Observed"). However, we have not explicitly shown the total water balance including the GRACE TWS term. We will provide more information about the overall water balance and add plots to the Supplementary Information. Note that the overall "observed" water balance (calculated using GRACE, GRDC and the precipitation datasets) does not close, which supports the suggestion of a low bias in the precipitation.

ii) Biases, broadly similar to those seen in JULES, have also been found in other models – see references in our Introduction (particularly page 3, line 12 and the preceding paragraph). This supports the idea that at least some of the bias is due to the input data, but does not rule out the possibility that the JULES model is a source of systematic bias. While we show that using the scaled precipitation does improve the representation of SWE in JULES, it does not completely remove the biases in all basins and for all driving data sets. Some of this remaining bias is likely due to model uncertainty. We will add more discussion of this to the text.

iii) We agree that issues related to energy fluxes and thermal state can also impact the modelled hydrology. While that is potentially a broad topic, we feel that in the context of our work these effects are less significant for the relatively high-latitude areas that form most of our study area. Winter conditions are severe in these areas and the land surface (including near-surface soil) is frozen for several months of every year. In terms of the development of the snowpack, the thermal state of the ground might be expected to play a more important role in the shoulder seasons (and at lower latitudes) – e.g. warm ground might slightly delay the deepening of the snowpack. However this effect will be relatively small in these very cold regions that reliably freeze every year and will have little impact on the seasonal maximum SWE in comparison to the effects of biased winter precipitation inputs.

In the following, we will respond to selected specific comments. All others we consider to be useful advice and will implement changes and expand explanations as recommended.

**1  Abstract**

**Line 17**  Re our emphasis on and comparison of maximum SWE and river discharge in the abstract. We are slightly unsure as to the meaning of this question. We emphasise these quantities because they are central to our study and closely related. If the question is why *maximum* SWE and discharge, as discussed in the manuscript we expect the maximum SWE (i.e. close to end of accumulation season)

to be closely related to discharge, particularly the spring flood which is such a large component of the annual discharge cycle in these rivers.

**2 Introduction**

**Page 9, Line 32 – Page 10, Line 1:** The model does not simulate a transition state — the precipitation is either rainfall or snowfall. It could be possible to have a transitional range of temperatures but this has not been implemented in JULES. The literature on this subject suggests that the details of the transition are themselves highly uncertain and vary with atmospheric conditions.

**Page 13, Line 4 – 5:** The SWE from JULES is less sharply peaked but we do not know why. It is possible that this is a feature of the wintertime meteorological driving data (possibly a particular bias towards missing smaller events towards the end of the accumulation season) and/or of how the model responds to those particular conditions (possibly it tends to start to melt slightly too early).

**Page 17, Line 28:** Undercatch correction is a known issue for rainfall as well as snowfall, therefore we surmise that it may also be an issue in the summer in these basins. See for example [1] and [2].

**Page 19, Line 1 – 2:** We did not save the interception values from the model. However, this would be instructive, so we could re-run the model and allow it to output the interception to add to the discussion.

**3 Discussion**

We selected the maximum SWE as the variable for evaluating the model as this is most closely linked to the water that is available for the spring melt and therefore the overall water balance. The starting date of snow accumulation is less important for seasonal/annual water balance calculations and is possibly more dependent on temperature regime than available precipitation.

The effect of antecedent water storage on the SWE will likely affect the start of accumulation, as it will change the heat capacity and therefore whether the snow melts or stays frozen. However, we think this is likely to have a small effect – see discussion under 'Abstract' above.

**4 Figures**

**Figure 2:** We define the cold-season accumulation period to be October-February. This is defined based on two metrics

1. The change in GRACE TWS ($\Delta$TWS) must be positive
2. The evaporation must be small

See Fig S4 for the fluxes. The reason that we do not include March is that the inter-annual variability is quite high, and in some years and some basins, the metrics above are violated in March. We chose a conservative accumulation period that would apply over all years and basins.

**References**

[1] Jennifer C. Adam and Dennis P. Lettenmaier. Adjustment of global gridded precipitation for systematic bias. *Journal of Geophysical Research: Atmospheres*, 108(D9), 2003. doi:10.1029/2002JD002499.

[2] Jennifer C. Adam, Elizabeth A. Clark, Dennis P. Lettenmaier, and Eric F. Wood. Correction of global precipitation products for orographic effects. *Journal of Climate*, 19(1):15–38, 2006. doi:10.1175/JCLI3604.1.

---

## Author Comment (AC2) · 4 Oct 2019

**Response to Referee 2:**
**Using GRACE to derive corrections to precipitation data sets and improve modelled snow mass at high latitudes**

Emma L. Robinson        Douglas B. Clark

We thank the reviewer for their comments and suggestions. In response to the questions raised:

***Page 4, Line 31-32:*** We have compared the basin areas of our $1°$ resolution river basins with the basin areas quoted in the GRDC dataset (these are the areas draining to the station nearest to the basin outflow, see Table 1 below). The difference is a few percent for each basin. The biggest difference is in the Lena, where our basin area is 6% smaller than the GRDC area.

| Basin | GRDC basin area (km$^2$) | TRIP basin area (km$^2$) | Difference |
|---|---|---|---|
| Yenisei | 2440000 | 2492975 | +2% |
| Ob | 2949998 | 2920094 | -1% |
| Lena | 2460000 | 2315590 | -6% |
| Mackenzie | 1660000 | 1721400 | +4% |

Table 1: Areas of river basins draining to the GRDC station, as given by the GRDC data set [1] and by the river basins from the $1°$ TRIP river network [2]. The final column shows the percentage difference between the TRIP and the GRDC areas.

***Page 6, Section 2.2:*** GRACE RL05 is the set of spherical harmonic solutions. We follow the recommendation of [3] and use the mean of the the three available solutions to effectively reduce the uncertainty, rather than using the single mascon solution.

***Page 7, Section 2.4:*** We decided that it would be better to use a best estimate of evaporation in these basins, rather than using something that is known to be wrong (zero cold season evaporation). GLEAM is a model output, but it is well established and based on gridded observational data sets that are independent of the data that we are investigating in this analysis.

In developing this analysis, we also investigated the use of the LandFlux-EVAL dataset (a synthesis of diagnostic, land-surface model, global hydrological model and reanalysis ET estimates) [4], as well as using JULES output ET. LandFlux-EVAL only has a few years in common with the GRACE timeseries, so could not be used for the whole analysis. We did not want to use JULES output as this is not independent of the precipitation data that we are trying to investigate.

We will add some discussion of this, as well as the inherent uncertainties in the observational products to the revised text.

***Page 11, Equation 5-6:*** These formulations are equivalent. We have chosen to use the one that demonstrates the way that we have combined the other products to obtain an estimate of precipitation, so would prefer to keep this as is.

***Page 12, Section 4.3:*** Yes, it would be a useful exercise to see the effect of using this scaling for the whole year. Of course, it should not be relied upon as a definitive scaling for the summer precipitation, and one would expect that the underestimation in summer precipitation results from different processes. But it would be interesting to see the effect of using our cold season scaling for the whole year and it would give an indication of the size of the required scaling for the summer precipitation that would result in better discharge simulations. We will do this experiment and will include the results in the supplementary information.

***Page 17, Line 19:*** Yes, we will do this.

**References**

[1] GRDC. The global runoff data centre, 2014. URL `https://www.bafg.de/GRDC`. 56068, Koblenz, Germany. Accessed: 2014-09-01.

[2] Taikan Oki and Y. C. Sud. Design of total runoff integrating pathways (trip)—a global river channel network. *Earth Interactions*, 2(1):1–37, 1998. doi:10.1175/1087-3562(1998)002<0001:DOTRIP>2.3.CO;2.

[3] C. Sakumura, S. Bettadpur, and S. Bruinsma. Ensemble prediction and intercomparison analysis of grace time-variable gravity field models. *Geophys. Res. Lett.*, 41(5):1389–1397, March 2014. ISSN 0094-8276. doi:10.1002/2013gl058632.

[4] B. Mueller, M. Hirschi, C. Jimenez, P. Ciais, P. A. Dirmeyer, A. J. Dolman, J. B. Fisher, M. Jung, F. Ludwig, F. Maignan, D. G. Miralles, M. F. McCabe, M. Reichstein, J. Sheffield, K. Wang, E. F. Wood, Y. Zhang, and S. I. Seneviratne. Benchmark products for land evapotranspiration: Landflux-eval multi-data set synthesis. *Hydrology and Earth System Sciences*, 17(10):3707–3720, 2013. doi:10.5194/hess-17-3707-2013.

---

## Author Response (AR1)

**Point-by-point response to Referee 1:**
**Using GRACE to derive corrections to precipitation data sets and improve modelled snow mass at high latitudes**

Emma L. Robinson        Douglas B. Clark

We thank the reviewer for their comments and suggestions, we have revised the manuscript accordingly. We provide a point-by-point response to the reviewer, noting that some of this is repeated from our response during an earlier phase of the review process. In response to the main concerns:

i) In the course of the analysis, we have indeed investigated the overall water balance, and it suggests that there is not enough precipitation in any of the data sets to support the observed river flows. We originally included the GRDC basin discharge and GLEAM evapotranspiration in Fig. S10 in the Supplementary Information (labelled "Observed"). We have now included a plot (Fig. S10), which shows that the overall "observed" annual water balance does not close when the precipitation data sets in this study are used. Although in itself this is insufficient to say which terms are in error, the rest of the manuscript develops the proposition that there is a considerable error in the precipitation term, at least. We have also added some text to the Discussion section of the manuscript.

ii) Biases, broadly similar to those seen in JULES, have also been found in other models – see references in our Introduction (particularly page 3, line 12 and the preceding paragraph). This supports the idea that at least some of the bias is due to the input data, but does not rule out the possibility that the JULES model is a source of systematic bias. While we show that using the scaled precipitation does improve the representation of SWE in JULES, it does not completely remove the biases in all basins and for all driving data sets. Some of this remaining bias is likely due to model uncertainty, but a thorough examination of model uncertainty as beyond the scope of this study. We have added more discussion of this to the text, including pointers to studies of model uncertainty (see the Discussion section).

iii) We agree that issues related to energy fluxes and thermal state can also impact the modelled hydrology. While that is potentially a broad topic, we feel that in the context of our work these effects are less significant for the relatively high-latitude areas that form most of our study area. Winter conditions are severe in these areas and the land surface (including near-surface soil) is frozen for several months of every year. In terms of the development of the snowpack, the thermal state of the ground, melt events and other fluxes of energy and might be expected to play a more important role in the shoulder seasons (and at lower latitudes) – e.g. warm ground might slightly delay the deepening of the snowpack. However this effect will be relatively small in these very cold regions that reliably freeze every year and will have little impact on the seasonal maximum SWE in comparison to the effects of biased winter precipitation inputs.

**1   Abstract**

**Line 15**  Done

**Line 17** Re our emphasis on and comparison of maximum SWE and river discharge in the abstract. We are slightly unsure as to the meaning of this question. We emphasise these quantities because they are central to our study and closely related. If the question is why *maximum* SWE and discharge, as discussed in the manuscript we expect the maximum SWE (i.e. close to end of accumulation season) to be closely related to discharge, particularly the spring flood which is such a large component of the annual discharge cycle in these rivers. We have added a sentence to Section 2.5.3 to reiterate this.

**2 Introduction**

**Page 2, Line 1-7:** Done

**Page 2, Line 8:** Rephrased

**Page 2, Line 18:** Done

**Page 2, Line 22-24:** Rephrased

**Page 2, Line 27:** "the impact" is the impact of climate change on SWE. Rewritten to clarify.

**Page 3, Line 2:** Done

**Page 3, Line 24:** Done

**Page 3, Line 26-28:** Rephrased

**Page 3, Line 30-31:** Rephrased

**Page 4, Line 10:** Deleted (this was a stray from earlier in the paragraph)

**Page 4, Line 17:** Rephrased

**Page 4, Line 19-22:** Rewritten to remove results

**Page 8, Line 13:** Done

**Page 9, Line 32 – Page 10, Line 1:** The model does not simulate a transition state — the precipitation is either rainfall or snowfall. It could be possible to have a transitional range of temperatures but this has not been implemented in JULES. The literature on this subject suggests that the details of the transition are themselves highly uncertain and vary with atmospheric conditions.

**Page 10, Eq. 1:** Yes, this should be (i, b).

**Page 10, Eq. 2:** $M_S$ is the mean over all basins. We have revised the text to make this clear.

**Page 11, Eq. 5:** Yes, have added units

**Page 13, Line 4 – 5:** The SWE from JULES is less sharply peaked but we do not know why. It is possible that this is a feature of the wintertime meteorological driving data (possibly a particular bias towards missing smaller events towards the end of the accumulation season) and/or of how the model responds to those particular conditions (possibly it tends to start to melt slightly too early). We have added this to the text

**Page 13, Line 8-9:** Rephrased

**Page 15, Line 7-8:** Have moved the Figure reference to here

**Page 15, Line 17:** Done

**Page 17, Line 13-15:** Rephrased

**Page 17, Line 28:** Undercatch correction is a known issue for rainfall as well as snowfall, therefore we surmise that it may also be an issue in the summer in these basins. See for example [1] and [2]. We have added this to the text.

**Page 19, Line 1 – 2:** We have retrieved the interception (and other evaporation components) from the model runs. This supports our assertion (CRUNCEP has much higher interception than the other data sets). We have added to the text.

**3   Discussion**

We selected the maximum SWE as the variable for evaluating the model as this is most closely linked to the water that is available for the spring melt and therefore the overall water balance. The starting date of snow accumulation is less important for seasonal/annual water balance calculations and is possibly more dependent on temperature regime than available precipitation. We have added some text at the point where seasonal maximum SWE is first introduced (Section 2.5.3).

The effect of antecedent water storage on the SWE will likely affect the start of accumulation, as it will change the heat capacity and therefore whether the snow melts or stays frozen. However, we think this is likely to have a small effect – see discussion under 'Abstract' above.

**4   Figures**

**Figure 1:** Added axis label.

**Figure 2:** Added axis label.

We define the cold-season accumulation period to be October-February based on two metrics

1. The change in GRACE TWS ($\Delta$TWS) must be positive
2. The evaporation must be small

See Fig. S4 for the fluxes. The reason that we do not include March is that the inter-annual variability is quite high, and in some years and some basins, the metrics above are violated in March. We chose a conservative accumulation period that would apply over all years and basins.

We have added this to the text.

**Figure 3:** Added axis label.

**Figure 6&7:** Added axis labels.

**Figure 8:** Fixed axis label.

**5   Figures**

**Figure S1, S2, S9:** Fixed axis labels.

**Figure S4:** Fixed caption.

**Point-by-point response to Referee 2:**
**Using GRACE to derive corrections to precipitation data sets and improve modelled snow mass at high latitudes**

Emma L. Robinson          Douglas B. Clark

We thank the reviewer for their comments and suggestions, we have revised the manuscript accordingly. We provide a point-by-point response to the reviewer, noting that some of this is repeated from our response during an earlier phase of the review process. In response to the questions raised:

***Page 4, Line 31-32:*** Throughout we have used the area of the basin calculated by integrating all of the $1°$ grid squares that drain to the grid square containing the gauging station. We have compared the basin areas of our $1°$ resolution river basins with the basin areas quoted in the GRDC dataset (these are the areas draining to the station nearest to the basin outflow, see Table 1 below). The differences range between -6% to +4%, which would result in a bias of between +6% to -4% in the estimate of basin discharge, compared to scaling with the GRDC basin areas. However, the discharge only contributes a small proportion of the GRACE-derived precipitation estimate (which is dominated by the GRACE $\Delta$TWS during winter – see Fig. S4), and the small percentage change in discharge results in little difference to the precipitation estimates (and therefore to the scale factors). Further, the uncertainty in the GRDC basin areas is possibly quite large. In view of this we used the $1°$ resolution river basin maps and areas throughout.

| Basin | GRDC basin area (km$^2$) | TRIP basin area (km$^2$) | Difference |
|-------|--------------------------|--------------------------|------------|
| Yenisei | 2440000 | 2492975 | +2% |
| Ob | 2949998 | 2920094 | -1% |
| Lena | 2460000 | 2315590 | -6% |
| Mackenzie | 1660000 | 1721400 | +4% |

Table 1: Areas of river basins draining to the GRDC station, as given by the GRDC data set [1] and by the river basins from the $1°$ TRIP river network [2]. The final column shows the percentage difference between the TRIP and the GRDC areas.

***Page 6, Section 2.2:*** We used the GRACE RL05 spherical harmonic solutions. We follow the recommendation of [3] and use the mean of the the three available solutions to effectively reduce the uncertainty, rather than using the single mascon solution. We have reiterated this in the text.

***Page 7, Section 2.4:*** We decided that it would be better to use a best estimate of evaporation in these basins, rather than using something that is known to be wrong (zero cold season evaporation). GLEAM is a model output, but it is well established and based on gridded observational data sets that are independent of the data that we are investigating in this analysis.

In developing this analysis, we also investigated the use of the LandFlux-EVAL dataset (a synthesis of diagnostic, land-surface model, global hydrological model and reanalysis ET estimates) [4], as well as using JULES output ET. LandFlux-EVAL only has a few years in common with the GRACE

timeseries, so could not be used for the whole analysis. We did not want to use JULES output as this is not independent of the precipitation data that we are trying to investigate.

We have added some discussion of this, as well as the inherent uncertainties in the observational products to the revised text.

***Page 11, Equation 5-6:*** Yes, this had the wrong signs. These have been fixed.

***Page 12, Section 4.3:*** Yes, we have carried out this exercise and it has been instructive. We have added a figure to the SI (Fig. S12) which shows that it has a significant effect on the basin discharge for CRUNCEP and PGF (less so for WFDEI, as the scaling was small), but with little effect on the SWE. We have added text to the Discussion section.

***Page 17, Line 19:*** Done

**2 Supplementary information**

We added two figures:

**Figure S10** a summary of the annual water balance

**Figure S12** a repeat of Figure 5, but with an extra column showing the basin discharge when the scale factors are applied to the whole year.

[revised manuscript text omitted]

---

## Author Response (AR2)

**Production upload:**
**Using GRACE to derive corrections to precipitation data sets and improve modelled snow mass at high latitudes**

Emma L. Robinson          Douglas B. Clark

Some trivial changes were made to the text before submission for production:

- Since submitting the paper, the name of our institution has changed to "UK Centre for Ecology & Hydrology", so we have changed our affiliation accordingly.

- We have updated the acknowledgements.

- We have moved the figures to the end of the text.